# The Use of Lidocaine-Infused Castration Bands to Castrate Beef–Dairy Calves and Its Effect on Animal Welfare and Performance

**DOI:** 10.3390/ani15040538

**Published:** 2025-02-13

**Authors:** Madeline R. Mancke, Eduarda M. Bortoluzzi, Payton Dahmer, Brad J. White

**Affiliations:** 1Beef Cattle Institute, Kansas State University, Manhattan, KS 66506, USA; mmancke@vet.k-state.edu (M.R.M.); bwhite@vet.k-state.edu (B.J.W.); 2Department of Anatomy and Physiology, Kansas State University, Manhattan, KS 66506, USA; 3Department of Animal Sciences and Industry, Kansas State University, Manhattan, KS 66506, USA; dahmerp@ksu.edu

**Keywords:** banding, castration, cattle, lidocaine, welfare

## Abstract

Castration is a common practice performed in the beef industry to manage male calves. Animal welfare concerns exist following castration due to the presence of pain and distress. Banding is one common castration method; however, it causes both acute and chronic pain. A lidocaine-infused castration band (LLB) was formulated to reduce pain. Therefore, the present study’s focus was to determine possible benefits of using an LLB on young calves (less than 2 weeks old). Performance (body weight, average daily gain, feed conversion), behavior measures, and blood biomarkers indicative of pain and distress demonstrated that banding with an LLB was comparable, if not outperformed, to a standard band. Calves banded with a LLB showed greater weight gain recovery compared to calves banded with a standard band. Those calves also performed more movements that were indicative of less pain being present (increased lying movements, decreased number of wound licks) around the scrotal sloughing time. Banding calves with an LLB allows producers to benefit from the efficiency of the procedure while addressing the animal welfare concerns around banded castration.

## 1. Introduction

The practice of crossbreeding beef x dairy cattle has grown significantly in the U.S. The use of sexed beef semen in dairy herds is attributed to an effort to increase the value of surplus calves. In 2021, a survey conducted with 40 Wisconsin dairy farms indicated that, on average, 454 beef–dairy cross calves were born per year per dairy farm; of those, we could consider 89% to be males [1,2,3]. If we extrapolate this to all farms in the U.S. with milk cows in 2022 (n = 36,024) [4] starting to use beef semen, there is a possibility that 16,354,896 male beef x dairy calves will be born and supplied to the beef industry per year. Due to differences and a lack of research regarding beef–dairy cross calf management, as well as the implications for animal welfare compared to traditional beef calf management, more research on management strategies is warranted for these calves entering the beef supply chain.

A common management procedure that is performed in both the dairy and beef industries is castration. Castration is recommended to be performed early in life, between birth and 3 months of age [5]. Male beef calves are castrated to reduce aggressive behavior, avoid pregnancies in cattle, and enhance carcass quality traits [6]. In a 2010 survey, elastrator rubber rings were the most commonly used non-surgical castration method in calves less than 90 kg (44%) [7]. Banding consists of placing a latex rubber band around the neck of the scrotum, entrapping the spermatic cords, and restricting blood supply to the testicles and scrotum, causing death and necrosis of these tissues and eventual “casting”, i.e., when necrotic tissues distal to the band fall off. Significant welfare concerns associated with this procedure are pain and distress due to the acute and chronic pain involved [8,9]. Previous studies reported that calves castrated with bands presented reduced performance traits and increased chronic pain and discomfort-related behaviors compared to those treated by surgical castration [10].

Lidocaine is a common analgesic, with many administrative routes being proven valuable in aiding in acute and chronic pain [11]. To aid in local pain mitigation, a lidocaine-infused castration band (LidoBand™, AVL/Solvet Ltd., Calgary, AB, Canada) was developed to slow-release therapeutic doses of lidocaine in the scrotum tissue under the band. It has been shown that calves banded with a lidocaine-infused castration band had tissue levels of lidocaine that approached 50% effective concentration (EC50) and exceeded 95% effective concentration (EC95) at 2 and 72 h, respectively, following castration, remaining above those levels for at least 28 days post castration [12]. In lambs banded with a lidocaine-infused castration band, lidocaine concentration in tissue at castration and tail docking sites met or exceeded the EC50 for at least 21–28 days and showed local anesthesia at the tail site for at least 3 days compared to a control band, determined by the electrostimulation data [13]. A study comparison between control bands, plus a local subcutaneous (SC) lidocaine injection and lidocaine-infused bands without a lidocaine injection used as castration methods, reported that lidocaine-infused bands calves displayed less pain response upon in situ electrostimulation than the control-bands-plus-one-lidocaine-injection calves at 7 days post banding [14]. However, no studies compared welfare outcomes based on basic health and function, affective states (pain and discomfort), and the natural living of calves castrated using lidocaine-infused bands for longer than 7 days.

The objectives of this study were to compare the measures of welfare (basic health and function and affective states (pain and discomfort)) of calves banded with Lidoband™ compared to calves banded with an identical standard band with no pain mitigation over a 49-day post castration period. We hypothesized that calves banded with Lidoband™ will show greater measures of basic health and function and decreased pain and discomfort, indicating the benefits of banding with a lidocaine-infused castration band.

## 2. Materials and Methods

All procedures described within this study were approved by the Animal Care and Use Committee of Kansas State University (IACUC-4923-AS&I). Due to the nature of a pilot study, the sample size was determined by previous studies with similar study designs [10,14].

A total of 26 intact male Angus Holstein (beef–dairy) cross calves, less than 2 weeks old and weighing between 36 and 54 kg, were used to conduct this study at the Sheep and Meat Goat Unit at Kansas State University. This experiment was conducted as a blinded, randomized controlled trial, with the treatment groups being (1) lidocaine-infused castration band (Lidoband™; LLB; n = 13) and an identical standard castration band with no pain mitigation (CONT; n = 13). Calves were randomized to treatment group A or B (to ensure blinding) by their study identification number using a random number generator on Excel (version 3.12.1., Microsoft, Redmond, WA, USA). Individual calf was the experimental and observational unit. All castrations were performed by one unblinded veterinarian. Banding occurred on 11 October 2023, which notes Day 0 of the study, and calves were then followed for 49 days, with the intention being to observe scrotal sloughing for the majority of the calves.

All calves were transported approximately 300 miles on 9 October 2023 from a commercial calf-ranch in Southwest Kansas. Upon arrival at the study facility, all calves were weighed, given metaphylaxis (tulathromycin), a tetanus toxoid, and uniquely identified with an ear tag with their corresponding study identification number. Calves were given an oral coccidiostat (Corid (amprolium), Huvepharma, Peachtree City, GA, USA) once daily for 5 days after arrival. A two-day acclimation period was given to the calves prior to the start of the study. All calves had a health check performed by a veterinarian and were deemed clinically healthy prior to the study start date and at the conclusion of the study. During the experiment, calves were individually housed in raised pens (1.5 × 2.4 m) in an indoor facility, where the pens were cleaned twice daily. Sheep panels separated the individual pens, allowing calves to still see other calves, and neighboring calves were still able to touch and interact with each other. There was no roof over the pens. Each pen had an individual water bucket, feed bucket, and milk bottle holder. Calves were fed 2 L of milk replacer (Herd Maker^®^ PB Milk Replacer, Land O’Lakes^®^, Arden Hills, MN, USA; 22% crude protein, 15% crude fat, 0.4% crude fiber) twice daily, and fresh water was always provided. A commercially available starter ration (Calf Grower B-68, Mid Kansas Coop, Manhattan, KS, USA; 14% crude protein, 2% crude fat, 7% crude fiber; 89% DM) was provided for the duration of the study. Health checks performed by experienced personnel were conducted on all calves twice daily at feeding, and band retention was also checked. All animal caretakers and analyzers were blinded to the treatment group. All outcome measures were collected on calves in their individual pen, except body weights, which were measured by a stationary calibrated livestock scale located next to the pens. Data collection happened after morning feeding, at approximately hour 08:00, in the order of bodyweight, blood draws, banding site pictures, and video monitoring (15 min post human interaction).

Feed and milk intake was recorded once and twice daily, respectively, for 49 days by measuring the amount offered minus the amount refused (unconsumed). Body weight was measured at arrival and on days 0, 7, 14, 21, 28, 35, 42 and 49 to calculate the average daily gain (ADG). With daily feed and milk intake being measured on a dry matter basis, as were the weekly body weights, a gain to feed ratio (G:F) was calculated to determine feed conversion by dividing the weight gained by the feed consumed.

Banding site pictures were taken prior to banding, at the time of band castration, and on days 1, 2, 7, 14, 21, 28, 35, 42, and 49 following banding. Using a handheld camera (Canon EOS Rebel T3i, Ohta-ku, Tokyo, Japan), photographs were taken approximately 30 cm from the caudal end of the calves while they were standing, with the entirety of the scrotal tissue and band being visible. Banding site pictures were scored post-study by a single, blinded, trained observer, and scores were given to each banding site picture based on a published scored system from 1 to 5 [8]. The present study’s scoring rubric is shown in Figure 1. While checking for band retention, the date of scrotal sloughing was recorded, if applicable.

While calves were laying down and restrained by personnel within their individual pens, approximately 10 mL of blood was collected from the jugular vein using 10 mL syringes and a 12G × 1″ needles at the following timepoints: days −1, 0 (band castration), hours 2 and 6, and days 1, 2, 7, 14, 21, 28, 35, and 42, at approximately hour 08:00, after body weights were calculated and before the banding site pictures were taken. Samples were immediately placed on ice after collection. Five milliliters were added to a blood collection tube intended for cortisol analysis (6 mL Greiner Bio-One VACUETTE™ Lithium Heparin Blood Collection Tube, Greiner Bio-One North America Inc., Monroe, NC, USA) and 5 mL were added to a blood collection tube intended for Substance P analysis (6 mL Greiner Bio-One VACUETTE™ Lithium Heparin Blood Collection Tube, Greiner Bio-One North America Inc., Monroe, NC, USA). Forty-eight hours prior to the blood collection timepoints, benzamidine hydrochloride (final concentration 1 mM) was added to the EDTA blood tubes for Substance P (SP) analysis. The Benzamidine HCl solution was prepared in a clean lab wearing proper PPE including gloves and a lab coat. Briefly, 156.6 mg Benzamidine HCl was weighed out and added to 10 mL MilliQ ultrapure water in a sterile 15 mL conical. Solution was vortexed thoroughly and until dissolved. Blood tube caps were removed briefly to deliver the 60 uL of Benzamidine HCL solution to each tube using an Eppendorf M4 Repeat pipet (Eppendorf North America, Enfield, CT, USA). Tubes were recapped immediately and stored at 4 °C until use within 48 h. Blood was centrifuged at 1000× *g* for 15 min, and blood plasma was stored in a −80 °C freezer until analysis. Cortisol concentrations were determined through an enzyme-linked immunosorbent assay (ELISA; Caymen Chemical Cortisol ELISA kit item no. 500360, Ann Arbor, MI, USA). A total of 500 µL of each sample was aliquoted into a 10 mL glass tube. The pH was adjusted to 1.5 to 2 by adding HCL to each sample. Ethyl acetate (2 mL) was added to each sample and then vortexed. After the layers had separated, the ethyl acetate was transferred to a clean glass tube. The vortex and extraction process were repeated three times more, yielding about 8 mL ethyl acetate extract per sample. The ethyl acetate was evaporated by drying samples in a CentriVap set on Program 9 (37 °C) for about 2 h (until the samples were barely dry). The extract was dissolved in 500 µL of ELISA buffer (1X). The standard curve, ranging from 6.6 to 40,000 pg/mL, was created by diluting synthetic cortisol with ELISA buffer. An amount of 50 µL of each standard and sample was added to the appropriate wells in duplicate. A sample of 50 µL of the AChE cortisol tracer and cortisol ELISA monoclonal antibody were added to each well. The plates were then incubated overnight in the dark at 4 °C. Plates were washed at 300 µL for 5 cycles, and then 200 µL of Ellman’s Reagent was added to each well. Plates were then incubated in the dark for 90 min on a plate shaker. The plates were read at 405 nm. To determine the cortisol concentration of the samples, the raw data file was uploaded onto MyAssays Desktop software (version 7.0.211.1238 (200621)). Standard curves were plotted as a 4-parameter logistic curve. Samples were reanalyzed if the coefficient of variation (CV) was >16%.

A radioimmunoassay (RIA) was used to determine SP concentrations, ran in duplicate, using methods previously described [15]. By diluting synthetic SP (Pheonix Pharmaceuticals cat. No. 061-05) with RIA buffer (50 mM sodium phosphate dibasic heptahydrate, 13 mM disodium EDTA, 150 mM sodium chloride, 1 mM benzamidine hydrochloride, 0.1% gelatin, 0.02% sodium azide; pH 7.4), a standard curve, ranging from 20 to 128- pg/mL was created. In plain 12 × 75 mm conical bottom tubes, 100 µL of samples, standards, and quality controls (QCs) were aliquoted, followed by 100 µL of rabbit anti-SP primary antibody (1:20,000; Phoenix Pharmaceuticals cat. no H-061-05). With an RIA buffer, iodine-125-SP tracer (custom iodination by Revvity, NEX083000MC) was diluted to 20,000 cpm. An amount of 100 µL was then added to the samples, standards, and QCs. Samples were then covered and stored for 48 h at 4 °C. At the end of the incubation period, samples were placed on ice. A sample of 100 µL of normal rabbit plasma (1:80) and goat anti-rabbit secondary antibody (1:40; Jackson ImmunoResearch cat no. 111-005-003) were added to each tube. For 10 min, samples were incubated at room temperature, then placed back on ice. A sample of 100 µL of blank bovine plasma was added to the blank, zero, standard, and QC tubes. An amount of 1 mL of 12% polypropylene glycol in 0.85% sodium chloride was added to all the tubes. The supernatant was aspirated by centrifuging the samples at 3000 rpm for 30 min at 4 °C. For 1 min, tubes were counted on a PerkinElmer Wizard2 gamma counter. To determine the SP concentration of the samples, the raw data file was uploaded onto MyAssays Desktop software (version 7.0.211.1238 (200621)). Standard curves were plotted as a 4-parameter logistic curve. Samples were reanalyzed if the coefficient of variation (CV) was >16%.

Every study day (days −1 to 42), at hour 12:00, a human approach test was performed by a single, blinded observer. This was conducted by approaching each calf individually and recording whether the calf made any approach (physical movement) towards the human, and it was recorded as a yes (approach) or no (did not approach).

Three-axis accelerometers (Hobo Pendant ^®^ G Data logger, Onset Computer Corp., Bourne, MA, USA) were fastened to the hind left leg (metatarsophalangeal joint) on arrival. To attach the accelerometer, a small dab of tag cement (Kamar Adhesive) was placed on the hair of a calf’s leg and a four-inch cohesive bandage (AmerisourceBergen, Conshohocken, PA, USA) was loosely wrapped one time around the leg. The accelerometer was packed with a layer of foam and plastic film. Then, the accelerometer was fastened to the leg with a small dab of tag cement. A cohesive bandage was wrapped 6 more times around the leg to secure the accelerometer in place. Due to limited storage capacities, accelerometers were replaced twice over the duration of the study. Logger replacements happened on collection days while calves were laying down and restrained for blood collection. The time of removal and reposition were recorded, and, since loggers were launched for the day of replacement at midnight, it allowed the data to be stitched together without data loss. The axis position was recorded every 30 s for 24 h for 42 days. Axis position results were converted into binomial data (standing/lying) using a modified version of a published protocol [16]. Data were then summarized into total minutes standing, total minutes lying, number of stand and lie bouts (>60 s), and the average bout time per day. A bout was defined as a movement from standing to lying, or vice versa, and staying in that new position for at least one minute.

Mounted video cameras (Camera Bullet 4 Megapixel Motorized 2.8–12 mm lens, Geovision, Taiwan) were used to continuously record video data. Cameras were mounted at a 45-degree angle on a rail that was 10 feet above the pens, with 2–3 calves being able to be monitored per camera (Cameras 1, 2, 3, 4, 5 and 6 capture 3 pens per camera; Cameras 7, 8, 9 and 10 captured 2 calves per pen). The differences in the number of animals per camera were due to the number of cameras we had available. Nevertheless, the bullet cameras allow for zoom and lens adjustment automatically using the Geovision program; that way, only the pens of calves assigned to that specific camera would appear in the video. Videos samples were rendered 30 min per day on days −1, 0, 1, 2, 7, 14, 21, 28, 35, and 42, relative to band castration (at approximately hour 08:30). Calves were allowed 15 min of rest after human interaction (body weights, blood collection, banding site pictures) before the 30 min videos were recorded. Behavior annotation and scoring utilized a behavior-specialized software (e.g., The Observer XT version 16, Noldus Information Technology, Wageningen, The Netherlands) using an adapted ethogram, shown in Table 1. One single, blinded to treatment and timepoint, trained observer analyzed all the videos. The outcomes measured included the percentage of time standing and eating, standing and not eating, and lying down. Counts of tail flicks, wound licks, looking at flank, and foot stamping were also recorded.

Treatment groups were assigned as A and B at randomization by a researcher not involved in data collection or analysis. Researchers did not become unblinded to the treatment group until all data analysis and results were completed. All statistical analyses were run on R Studio (Version 2022.02.3, Boston, MA, USA). Banding site scores were binomialized to the following categories: before wound granulation (scores 1 and 2), and after wound granulation (scores 3 and 4). Wound granulation in the current study was defined as the occurrence of new tissue, pink or red in color, formed on the surface of the scrotal wound during the healing process. Banding site scores of 5 were not included in the data analysis due to only 2 calves presenting this score at the end of the study. Regarding the daily human approach test, days 0, 1, 2, 7, 14, 21, 28, 35, and 42 were excluded from the data analysis due to calves being approached by humans for body weight, banding site pictures, and blood collection prior to the 12:00 pm approach test. All data were assessed for normality using a Shapiro–Wilk test. Transformations were performed for video behavior scoring to achieve normality prior to statistical analysis. The percentage of time spent standing while not eating, the percentage of time spent standing while eating, and the percentage of time spent lying were square root + 1 transformed to achieve normality. Counts of tail flicking, looking at flank, and wound licking were log 10 + 1 transformed to achieve normality. Behavior data were transformed for statistical analysis and are presented as the least squares means and standard error without transformation, with a *p*-value from transformed values. For models accounting for repeated measures, fixed effects included treatment, timepoint, and treatment by timepoint interaction. For the models accounting for overall performance (final bodyweight at day 49 post castration and overall ADG and G:F over the 49 days), fixed effects included the treatment group only. A calf identification number was used as a random effect for all models (except overall performance) to account for repeated measures. An accelerometer number was used as a random effect for accelerometer data, as accelerometers were replaced twice during the study due to storage capabilities. Generalized linear mixed effects models (glmer function of lme4 package in R) were run to determine the possible differences between groups for banding site scores and approach tests. Linear mixed effects models (lme function of nlme package in R) were run to determine possible differences between the groups with all accelerometer data, video behavior scoring, performance data, and plasma biomarkers. Significance was established at *p* ≤ 0.05, with a tendency at 0.05 < *p* < 0.10, when biologically appropriate.

## 3. Results

Twenty-six single sourced beef-on-dairy calves were enrolled in this clinical trial. Thirteen calves were banded with a lidocaine-infused band (LLB) and 13 calves were banded with an identical standard band (CONT). The average initial weight of the LLB group was 43.8 kg, and the average initial weight of the CONT group was 46.5 kg (SEM 1.21, *p* = 0.13).

### 3.1. Body Weights, Average Daily Gain, Feed Conversion, and Banding Site Scores

The final body weight, overall ADG, and overall G:F were not significantly different between treatments (*p* > 0.05; Table 2). However, when data were analyzed weekly, a treatment by timepoint tendency was found in the ADG (*p* = 0.06) and a significant effect on G:F was found (*p* < 0.01). The LLB had a greater ADG than the CONT group in week 2 post castration (*p* < 0.05). A treatment effect was found in the G:F, with the LLB group having a greater feed conversion than the CONT group. No weekly treatment by time differences were found for bodyweights (*p* > 0.05). By day 49 post castration, 77% (10/13) of CONT calves and 62% (8/13) of LLB calves had cast their scrotal tissue. A timepoint effect for banding site scores was found, with the probability of the presence of granulating tissue increasing over time post castration, independent of the treatment (*p* < 0.05). No effects of treatment or treatment by time interactions were found (*p* > 0.05).

### 3.2. Blood Parameters

The Substance P RIA intra- and inter-assay variation was 6.19% and 11.84%, respectively. Cortisol ELISA intra- and inter-assay variation was 12.24% and 8.41%, respectively. There were no significant treatments by timepoint interactions for plasma cortisol (5726 pg/mL and 5331 pg/mL, SEM = 0.84; LLB and CONT, respectively) or Substance P concentration (231 pg/mL and 221 pg/mL, SEM = 0.71; LLB and CONT, respectively; *p* > 0.05). Plasma cortisol concentrations peaked immediately after band castration (~2.3 min post band castration) and decreased overtime regardless of the treatment group (timepoint *p* < 0.05).

### 3.3. Animal Behavior

No effects of treatment, timepoint, or their interaction were found for the daily human approach test (*p* > 0.05). A treatment by time interaction was found for the number of lying bouts, with LLB calves performing a greater number of lying bouts on days 35 to 41 post castration compared to CONT calves (*p* < 0.05; Table 3). The number of lying bouts decreased for both treatments post castration starting on day 2. The average standing time within a bout had a significant treatment by timepoint interaction, with CONT calves spending more time standing within a bout between days 18 and 22, day 30, and day 41 post castration (*p* < 0.05; Figure 2). When behavior was annotated for 30 min per day using videos 15 min after human presence, treatment by the timepoint effect was found on days 21 and 35 for time spent lying (%), with the LLB calves lying down for longer than the CONT calves (*p* < 0.05; Table 3). A significant treatment and timepoint effect were found for the number of wound licks. The CONT group licked their wounds more often than the LLB group (*p* < 0.05; Figure 3). The number of wound licks increased with time post castration for both the LLB and CONT groups. In addition, the number of times calves looked at their flank and the number of foot stomps performed also increased over time, regardless of treatment (*p* < 0.05; Figure 3).

## 4. Discussion

Castration pain has been demonstrated by several other studies looking at elements such as the acute pain in hours and days post procedure, as well as chronic pain weeks post procedure [6,8,9,10,17,18,19,20,21]. Researchers were able to demonstrate evidence of decreased affective states by the presence of pain and distress when assessing basic health and function using plasma biomarkers and banding site scores, as well as natural living by changes in behavior. The present study evaluated calves castrated with a lidocaine-infused band (Lidoband™) compared to calves banded with an identical standard band with no pain mitigation over a 49-day post castration period. Outcomes selected for this study aimed to assess both acute and chronic pain.

In the present study, there were no observed differences in overall performance outcomes. Conflicting differences in overall body weight and ADG were noted in previous research that compared band castration, knife castration, and sham. In one study, band castration and sham presented greater overall weaning weights compared to knife castration [8]. Another study reported lower weekly ADGs for band-castrated compared to knife-castrated calves [10]. However, each study had a different target population that included calves of similar ages but different breeds (extensive beef breeds, and individually housed dairy breeds, respectively). To the best of our knowledge, no studies tested differences in castration methods for young beef-on-dairy calves. Timepoint differences in weekly G:F, ADG, and cortisol suggest that acute pain was present immediately and during the first week post castration regardless of the treatment group. The reduction in feed conversion and weight gain during the first couple of weeks seen in the present study has been previously attributed to pain decrease, as well as findings of the decrease in testosterone in a population older than the one of the present study [22,23,24]. The pain during this period is likely the result of constriction and ischemia promoted by the bands. Humans who submitted to the use of torniquets during surgery described the pain of blood flow constriction as being dull, tight (with an aching sensation), and poorly localized [25]. However, the present study demonstrated differences and tendencies between treatments groups in G:F and ADG, respectively, when analyzed weekly. Calves in the LLB group had a greater weekly feed conversion and weekly ADG compared to CONT calves. The results indicate that LLB calves better coped with the acute pain caused by band-castration, which promoted better affective states. Nevertheless, no overall performance differences were noted. Cortisol and Substance P results found in the current study are similar to results from previous studies. The lack of differences in blood parameters (cortisol and Substance P) was previously described in rubber ring castration in young calves (one week old) compared to sham calves when assessing acute [9,26] and chronic pain [8]. Concerning cortisol, a similar lack of treatment differences was previously described in rubber-ring-castrated young calves (week old) compared to sham calves when assessing acute [9,26] and chronic pain [8]; however, cortisol peaked at castration time, indicating activation of the hypothalamic–pituitary–adrenal axis related to castration distress in both treatments. Although no treatment by time interactions were found, LLB calves seemed to have a greater G:F and ADG in the first two weeks post banding and lower cortisol immediately after band castration. A study with a greater number of calves is necessary to confirm the interactions between treatment and time favoring the LLB treatment group.

Differences in behavior occurred in a similar temporal window as wound granulation, where the band started cutting through the scrotal tissue to begin the sloughing process (around 20–40 days post castration). We hypothesized that this is the time where chronic pain occurs for banded calves. Previous studies mentioned that chronic pain might be present at 7 days post band castration until the testicles slough off [8], while others, attributed chronic pain to an analogy with humans and behavioral changes (e.g., abnormal postures and walking, licking lesion sites) that would likely indicate to producers and veterinarians evidence of long lasting pain [17]. Chronic pain or the long-term effects of castration were observed in the current study by timepoint effects observed for tail flicks, looking at flank, and foot stamping in the video analysis regardless of the treatment group. However, a treatment difference was found in the number of wound licks, with the LLB group licking their wounds less than the CONT group, indicating decreased pain in the scrotal area. Previous research demonstrated that wound-directed behaviors were associated with pain [10,17,18,27,28,29]. However, differences found between the treatment groups were very small, and previous research suggested that the size and less vascularized testicles in young calves could contribute to the lack of behavioral differences [8]. The accelerometer data showed a treatment by timepoint interaction with the number of lying bouts and time spent standing in one bout. The LLB calves spent less time standing in one bout and had more lying bouts within a day compared to the CONT calves. The movement between lying to standing position in cattle is performed using the hind limbs first and then the front limbs, causing a stretch of the abdominal muscles and skin. When a calf is castrated with a rubber band, the stretch of abdominal tissues is even more pronounced, which is believed to cause discomfort when lying to standing behavior is performed [9]. The LLB calves moved up and down more often, indicating less discomfort or pain in the scrotal area and suggesting that lidocaine in the band promoted some local analgesia. The time of scrotum casting in the present study was similar to a previous study where they noticed that about 70% of the calves casted by day 56, with only one calf (1/11) showing complete healing (score of 5) [10]. A limitation to the current study is the fact that not all calves sloughed their scrotal tissue and had completely healed banding sites by day 49. Further research is warranted to follow all banded calves through the time of scrotal sloughing and complete banding site healing to determine possible differences in behaviors or performance that the chronic pain of scrotal sloughing could cause. Thuer et al. (2007) showed that calves who were banded had a significantly greater pain response to palpation than control calves or calves castrated with a burdizzo after day 10 and up to 8 weeks [20], indicating the presence of chronic pain after banded castration.

Unfortunately, there are no set guidelines for times or intervals to score animal behavior. Other researchers assessing castration pain described their methodology of behavior scoring happening for 24 h once a week using a 5 min interval scan sampling [10]. Molony et al. (1995) behavior scored the first day for 3 h using 2 min intervals for the first 96 min and then 6 min intervals for the remaining 84 min [17]. In addition, when Meléndez et al. (2017) scored the behaviors of calves submitted to castration, their strategy was to conduct continuous behavior sampling between 2 and 4 h post castration and then continuously scan samples in 2 min intervals every 10 min from 8:00 to 14:00 h on days 1, 2, 3, and 5 post castration using a subset of five animals per treatment [9]. Later, Marti et al. (2017) used similar continuous scan sampling from 08:00 to 17:00 h on days 5, 13, 20 and 27 post castration to evaluate chronic pain [8]. Our strategy was to behavior score continuously for 30 min per day at the same time, following the exact same schedule on days −1, 0, 1, 2, 7, 14, 21, 28, 35, and 42 to maintain the homogeneity and consistency of the video data. Videos were rendered after morning feeding, bodyweight measures, blood collections, and banding site pictures were taken; then, animals were allowed a period of 15 min to return to natural behaviors without human interference. Our goal in allowing 15 min without human interaction was to diminish the masking of pain behaviors from human observation, since cattle can be stoic in the presence of humans. Nevertheless, the authors recognize that a 30 min behavior scoring may not represent behaviors occurring during the 24 h within a day. Behavior scoring present limitations due to amount of time necessary to score 24 h videos, which is clear not only from the present study but also from other research groups that have been previously mentioned [8,9,10,17]. New artificial intelligence behavior tools to decrease the amount of video hours needed to be watched by a researcher may provide a better tool to address the current challenge.

A limitation of this study is the small sample size. Our findings indicate that the use of lidocaine-infused castration bands may be a useful tool for producers to use for pain mitigation following banded castration. Collaboration with a calf ranch could lead to a commercial trial to assess performance metrics on a larger scale and increase external validity. Future work could include the use of Lidoband™ on different populations and sectors of the beef or dairy industry, giving valuable results to determine the population of cattle that would benefit the most from this product.

Results from the current study indicate that performance, behavior, and blood parameters are significantly influenced by banded castration. Performance differences observed early post castration and behavioral differences observed later post castration are indicative of Lidoband™ providing some alleviation from the acute and chronic pain that banding is known to cause. Regardless of treatment, acute and chronic pain are still present with banded castration and both short and long-term pain mitigation should be used to improve animal welfare.

## 5. Conclusions

This study demonstrates that the use of a lidocaine-infused castration band (LLB) offers significant advantages over standard banding for the welfare of young calves. While no major differences were observed in calf performance between the two treatment groups, the LLB-treated calves showed improved performance when the average daily gain (ADG) and feed efficiency (G:F) were analyzed weekly. Furthermore, behavioral indicators of pain, such as the number of wound licks and changes in standing and lying bouts, suggest that LLB treatment reduces pain and discomfort during the recovery period. These findings highlight the potential for LLB to mitigate the acute and chronic pain typically associated with band castration, making it a promising alternative that benefits both animal welfare and production outcomes.

## Figures and Tables

**Figure 1 animals-15-00538-f001:**
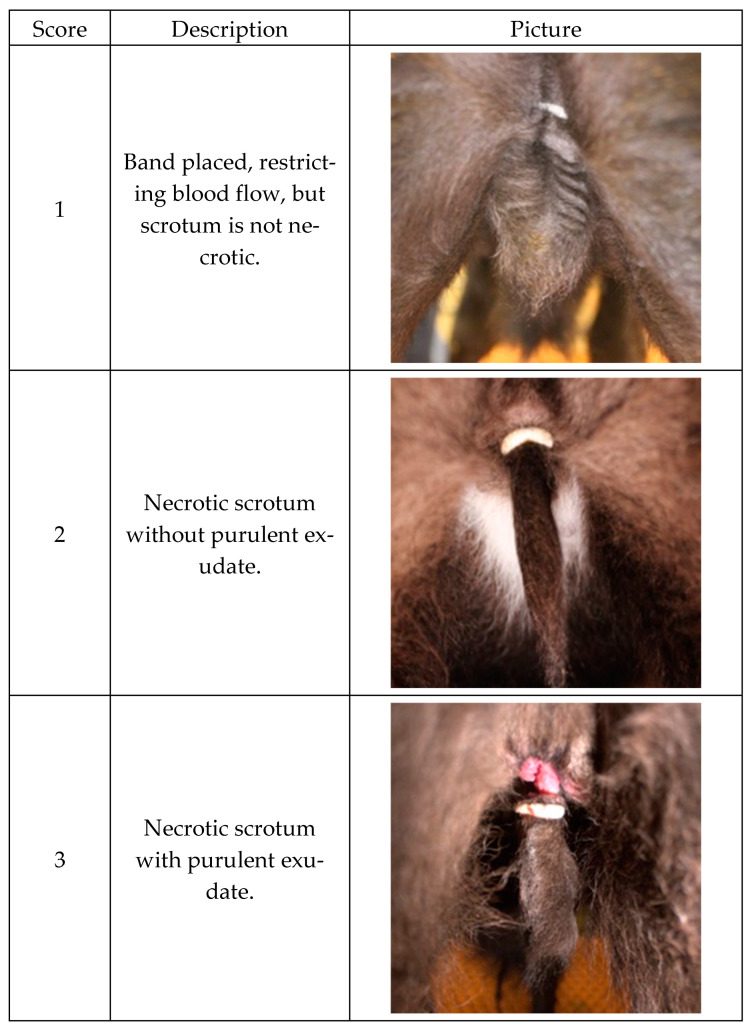
Five-point banding site scoring system used to evaluate the wound healing associated with Lidoband™ castration (LLB; n = 13) and standard band castration (CONT; n = 13). The score system used is a published scoring system from 1 to 5 [8], with pictures from the current study.

**Figure 2 animals-15-00538-f002:**
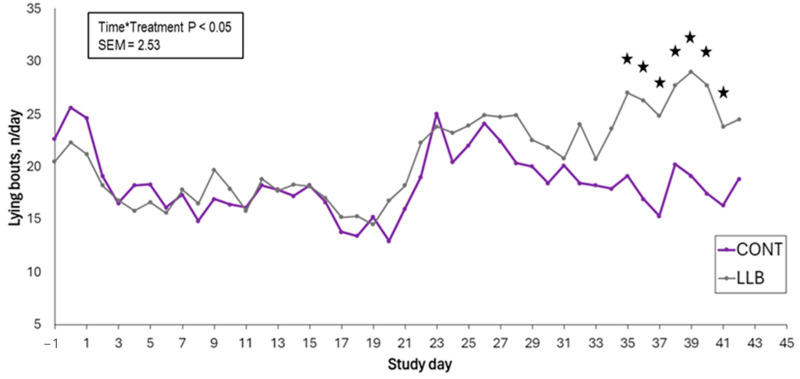
Average number of lying bouts per day of beef-on-dairy calves banded with a lidocaine-infused castration band (LLB; n = 13) and a standard band (CONT; n = 13), recorded by three-axis accelerometers. Stars in each study day denote differences among treatment groups (*p* < 0.05).

**Figure 3 animals-15-00538-f003:**
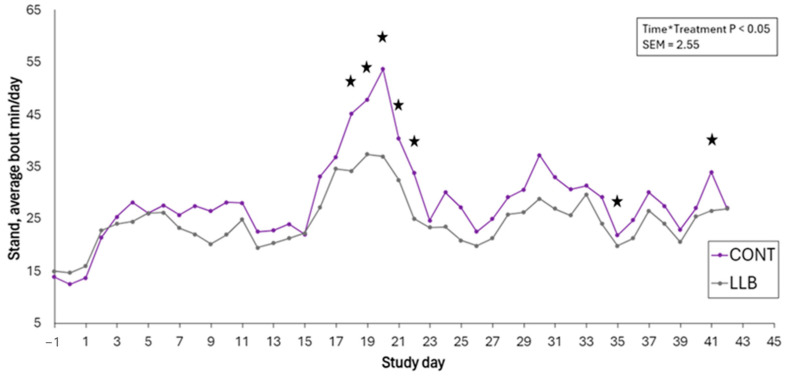
Average time (minutes) spent standing in a bout per day of beef-on-dairy calves banded with a lidocaine-infused castration band (LLB; n = 13) and a standard band (CONT; n = 13), recorded by three-axis accelerometers. Stars in each study day denote differences among treatment groups (*p* < 0.05).

**Table 1 animals-15-00538-t001:** Adapted ethogram used to evaluate behaviors associated with Lidoband™ castration (LLB; n = 13) and standard band castration (CONT; n = 13) through video monitoring.

Behavior	Description
Duration, regardless of chewing status	
Stand eating, %	Percent of time standing with head on feed bucket
Stand not eating, %	Percent of time standing with head outside feed bucket
Lying down, %	Percent time in sternal or lateral recumbency
Count	
Tail flicks, n	Number of forceful tail movements beyond the widest part of rump while standing
Wound licks, n	Number of times their head turned to lick the castration wound site while standing
Looks at flank, n	Number of times their head turned to look at the flank area
Foot stamping, n	Number of times one of the hind legs was lifted and forcefully placed on the ground or kicked outward while standing

**Table 2 animals-15-00538-t002:** Final body weight, overall average daily gain (ADG), and overall gain to feed (G:F) of beef-on-dairy calves banded with a lidocaine-infused castration band (LLB; n = 13) and a standard band (CONT; n = 13) and followed for 6 weeks post castration.

	Treatments			*p*-Value
	LLB	CONT	SEM	Treatment
Final body weight, kg	70.7	69.1	1.25	0.38
ADG, kg/d	0.51	0.49	0.02	0.49
G:F, kg	0.38	0.40	0.02	0.69

**Table 3 animals-15-00538-t003:** The least square means and standard error of behavioral parameters (duration and counts) obtained on day −1 (baseline), day 0 (immediately post castration), and days 1, 2, 7, 14, 21, 28, 35, and 42 post castration of beef-on-dairy cross calves band castrated with a standard band (CONTROL; n = 13) or with a lidocaine-loaded band (LLB; n = 13) scored in 30 min video samples.

	Treatment			*p*-Value		
*Behaviors*	CONT	LLB	SEM	Treatment	Timepoint	Treatment X Timepoint
Stand not eating, % ^1^	22.0	16.3	2.55	0.20	**<0.01**	0.06
Stand eating, % ^1^	2.4	1.0	0.63	0.87	**<0.05**	0.36
Lying down, % ^1^	73.4	81.2	2.69	0.33	**<0.01**	**<0.01**
Tail flicks, n ^2^	4.2	3.6	1.31	0.13	**<0.01**	0.11
Wound licks, n ^2^	1.8	1.6	0.41	**<0.05**	**<0.01**	0.10
Looks at flank, n ^2^	1.1	1.4	0.20	0.13	**<0.01**	0.35
Foot stamping, n ^2^	2.4	2.7	1.16	0.30	**<0.01**	0.19

The data presented are the least squares means and standard error without transformation and the *p*-value from transformed values. ^1^
*p*-values correspond to the ANOVA analysis using square root + 1 transformed data. ^2^
*p*-values correspond to the ANOVA analysis using log 10 + 1 transformed data.

## Data Availability

The original contributions presented in this study are included in the article. Further inquiries can be directed to the corresponding author.

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
