# Peer review of "The Use of Lidocaine-Infused Castration Bands to Castrate Beef–Dairy Calves and Its Effect on Animal Welfare and Performance"

_animals, 2025, doi:10.3390/ani15040538_

Round 1

Reviewer 1 Report

Comments and Suggestions for Authors

During the entire project assumptions about presence or absence of pain are very well made using the tests described. But never is there any mention of how long the Lidocaine persists in the band. 

Was the concentration of Lidocaine ever measured in the bands after the scrotum and band fell off. 

Author Response

Responses to reviewer 1:

Reviewer 1: “During the entire project assumptions about presence or absence of pain are very well made using the tests described. But never is there mention of how long the Lidocaine persists in the band. Was the concentration of Lidocaine ever measured in the bands after the scrotum and band fell off.

Authors: Thank you for your comment and response. As you stated, there was no mention of how long the lidocaine persisted in the band. This was not measured in the study, due to not fitting the study objectives of comparing measures of welfare (basic health and function; and affective states) of calves banded with Lidoband™ compared to calves banded with an identical standard band with no pain mitigation over 49-day post-castration period. This was an objective of the Saville et al. paper titled “Development and field validation of lidocaine-loaded castration bands for bovine pain management”. Their study showed that the lidocaine-infused bands delivered therapeutic quantities of lidocaine into scrotal tissues of cattle over a period of at least 7 days. However, there is no published literature that has shown lidocaine concentrations in the scrotal tissue and/or lidocaine-infused band at time of scrotal sloughing.

The authors would like to thank Reviewer 1 for the suggestion to improve our manuscript.

Reviewer 2 Report

Comments and Suggestions for Authors

Please review my comments in the file attached.

Author Response

Responses to reviewer:

Commented [R21]: “…around the neck of the scrotum…”

Authors: Thank you for this suggestion. This change was made on line 61 to state “Banding consists of placing a latex rubber band around the neck of the scrotum, entrapping the spermatic cords, and restricting blood supply to the testicles and scrotum causing death and necrosis of these tissues, and eventual “casting” i.e., when necrotic tissues distal to the band fall off.”

Commented [R22]: I suggest reviewing this statement and rewriting as “increased chronic pain and discomfort-related behaviors…”

Authors: Thank you for this suggestion. This change will be made on line 65 to state “Previous studies reported that calves castrated with bands presented reduced performance traits and increased chronic pain and discomfort-related behaviors compared to surgical castration [10].”

Commented [R23]: “…of up to 2-weeks old calves…”

Authors: Thank you for this suggestion. This change will be made on line 88 to state “The objectives of this study were to compare measures of welfare (basic health and function; and affective states (pain and discomfort)) of calves, less than 2 weeks of age, banded with Lidoband™ compared to calves banded with an identical standard band with no pain mitigation over a 49-day post-castration period.”

Commented [R24]: “Please provide the average BW and SD”

Authors: Thank you for this suggestion. The central measure of tendency and measure of variability between the two groups is stated in line 342-344 of the results section, since weight outcomes, even initial weight, were considered a result.

Commented [R25]: “Was bedding provided?”

Authors: Thank you for this question. No bedding was provided due to the nature of the flooring (rubber coated expanded metal), allowing for easy and thorough cleaning of the pens with drains localized just below and adjacent to their pens. We did not observe any reluctance on laying down/standing up as we observed those calves multiple times a day.

Commented [R26]: “Replace wound by “banding site””

Authors: Thank you for this suggestion. These changes will be made throughout the manuscript to state “banding site” instead of “wound site”.

Commented [R27]: “Same”

Authors: Thank you for this suggestion. Like stated above, these changes will be made throughout the manuscript to state “banding site pictures” instead of “wound pictures”.

Commented [R28]: “Why overnight? Please provide the reference for this lab analysis methodology.”

Authors: Thank you for this suggestion. This lab analysis methodology came directly from the kit instructions (Caymen Chemical Cortisol ELISA kit item no. 500360, Ann Arbor, MI), on line 194.

Commented [R29]: “This should go to L163.”

Authors: Thank you for this suggestion. This section will be moved to line 190 (163 prior to edits).

Commented [R210]: “Please provide a reference for this lab analysis methodology.”

Authors: Thank you for this suggestion. We had the CV number wrong; it is in fact a cut off >16%. The cutoff for CV at 16% was determined previously by our laboratory group (Coetzee and Bortoluzzi lab, at Kansas State university) to make certain that the assays were well run, and the data is precise and accurate. Our goal in adding this information is to be transparent of how the essays were ran.

Commented [R211]: “Standardize the terminology for this assessment (e.g., banding-site)”

Authors: Thank you for this suggestion. Like stated above, these changes will be made throughout the manuscript to state “banding-site pictures” instead of “wound pictures”.

Commented [R212]: “Does this included if animal was chewing?”

Response: Thank you for this question. This does include if the animal was chewing, if away from the feed bucket. Due to the angle of the cameras, and the calves eating with their faces away from the cameras, it was not feasible to determine chewing status, only the location of their head in comparison to the feed bucket. This will be added to Table 1 to further clarify.

Commented [R213]: “Please clarify.”

Authors: Thank you for this suggestion. As stated, researchers were blinded to treatment group until all analysis and results were completed. During analysis, researchers knew what calves were in group (treatment) A or B, but did not know what group (A or B) was LLB or CONT. Blinding was maintained throughout the entire process to decrease risk of bias (non-random error).

Commented [R214]: “Were they back transformed or not?”

Authors: Thank you for this question. As stated in our materials and methods and on footnotes of our tables and graphs. Data was not back transformed for presentation. The untransformed values were used for means and SEM and transformed values were used for P-values.

Commented [R215]: “Please ensure this is correct. Typically, the SEM presented is the transformed one.”

Authors: Thank you for your comment. Although many behavior data is presented as untransformed means and transformed SEM and P-values, we decided to present the means and SEM as untransformed values and P-values from transformed values. Our group discussed this approached with experts in data analysis and statisticians and we were advised that presenting untransformed means with transformed SEM should be avoided since they are in different scales, which can misrepresent variability and lead to misleading conclusions by readers. In addition, if we used the untransformed SEM as error bars in our graphs, those would not correctly represent the spread of untransformed data.

Commented [R216]: “…and a tendency at 0.05 < P < 0.10.”

Authors: Thank you for this suggestion. This sentence on line 337 will be restated to say “Significance was established at P ≤ 0.05 and a tendency at 0.05 < P < 0.10 when biologically appropriate.”

Commented [R217]: “…had greater ADG than….”

Authors: Thank you for this suggested change. This sentence on line 349 will be restated to say “The LLB group had a greater ADG than the CONT group in week 295 post castration (P <0.05)”.

Commented [R218[: “Standardize as previously mentioned.”

Authors: Thank you for this suggestion. Like stated above, these changes will be made throughout the manuscript to state “banding site scores” instead of “wound scores”.

Commented [R219]: “These 2 sentences should go to the methodology section.”

Authors: Thank you for this suggestion. However, the authors considered this being a result of the lab analysis, and will keep this information in the results section.

Commented [R220]: “Timepoint interaction”

Authors: Thank you for this suggestion. This change will be made to line 363 to state”…no significant treatment by timepoint interactions for plasma cortisol…”.

Commented [R221]: “Please correct this information on methodology (L270) as described in here”

Authors: Thank you for this suggestion. The corrections have been made.

Commented [R222]: “Please review this sentence and the references. Testosterone cannot be the cause for the feed conversion outcomes in less than 2-weeks od calves.”

Authors: Thank you for catching this. The authors agree that, at this young of an age, testosterone cannot be attributed to the weight gain of the calves. We will restate this sentence to say “…in the present study has been previously attributed to pain decrease, as well as findings of the decrease of testosterone in a population older than the one of the present study [22,23,24].”

Commented [R223]: “…time interaction…”

Authors: Thank you for the suggestion. This change will be made, and the sentence will state “…no treatment by time interactions were found…”.

Commented [R224]: “Please provide the amount of numerical differences observed.”

Authors: The authors were unsure of what the reviewer was referring to here. Nevertheless, if it is regarding discussion of numerical differences, the authors decided not to emphasize or discuss numerical differences without statistical significancy to reduce chance of Type I Error.

Commented [R225]: “I would say that the chronic pain starts when the skin cut-off. Please provide a more detailed information about when the pain might start using a reference”

Authors: Unfortunately, there is not set guidelines for when chronic pain starts. That is why we wrote that we hypothesized that that when the band cuts through tissue at the beginning of sloughing process is where the chronic pain starts. The International Association for the Study of Pain describes the following “Chronic pain is complex. But understanding its physical, emotional, and phycological and social components can help you communicate more effectively with health care providers, leading to better management and improved quality of life.” Nevertheless, we can add a sentence saying “Previous studies mentioned that chronic pain might be present at 7 days post band castration until the testicles sloughs off (Marti et al., 2017). While others, attributed chronic pain to an analogy with humans and behavioral changes (e.g. abnormal postures and walking, licking lesion site) that would likely indicate to producers and veterinarians evidence of long lasting pain (Molony er al., 1995).”

Commented [R226]: “Please clarify.”

Authors: Thank you for this suggestion. The sentence on line 488-490 will be added onto: “…through the time of scrotal sloughing and complete banding site healing to determine possible differences in behaviors or performance that the chronic pain of scrotal sloughing could cause.”

Commented [R227]: “What other studies showed post-day 48?”

Authors: Thank you for the question. A sentence will be added to the end of the paragraph, stating “Thuer et al. (2007) showed that calves banded had a significantly greater pain response to palpation than control calves or calves castrated with a burdizzo after day 10 up to 8 weeks [20], indicating the presence of chronic pain after banded castration.” We unfortunately could not follow calves for a longer period of time (until all sloughed) due to housing and financial constraints.

The authors would like to thank the reviewer for their comments and suggestions to make our manuscript better.

Reviewer 3 Report

Comments and Suggestions for Authors

General comments:

The manuscript titled “The use of Lidocaine-infused castration bands to castrate beef-dairy calves and its effect on animal welfare and performance”  aimed to compare behavior, growth performance, and blood parameters between calves castrated with a lidocaine-infused castration band versus traditional band. This is an important study to find a solution to mitigate the pain associated with castration. However, the manuscript in the current version lacks organization and is missing detailed information, which makes it difficult to assess the soundness of the presented data. While the study offers valuable insights, there are several areas where clarification and additional details are needed to improve the clarity and transparency of the methods and findings. Below, I address some of these concerns:

The manuscript states that a modified version of a previously published scoring system was used. However, the scoring system presented in this manuscript appears to be identical to the originally published system. I suggest removing the table to avoid redundancy and potential copyright concerns, as it has been directly taken from the previously published manuscript (Marti et al 2017).

Another main concern is that individual housing may prevent a clear observation of certain behaviors. The stress or fear caused by isolation could influence the results and make it difficult to interpret the true effects of Lidoband on calf behavior. Could you clarify how the housing conditions were managed to minimize this potential impact? For example, I am not sure how you were able to observe the hind legs of animals housed in individual housing. Could you please clarify the methods used for this observation?

The authors mentioned that data from the daily human approach test were excluded from the data analysis due to calves being approached by humans before the actual test. Why is this information still included in the manuscript? Please delete any mention of the human approach test throughout the manuscript and adjust your objective accordingly. Given that the animals were individually housed, I don't think the Human Approach Test is the right method to measure the effect of lidoband on the affective states of calves, as individually housed animals are likely to be more fearful.

I noticed that you presented only the final body weight (BW) of the calves, but BW was measured at different time points. Could you explain why only the final BW was included in the manuscript? Additionally, did you ensure that the initial BW was not significantly different between the two groups? Please provide the average initial BW for both groups to facilitate comparison.

Please provide additional details on blood collection, specifically the type of blood collection tubes used and the timing of blood sample collection in relation to other measurements.

How did you use mounted cameras to record the behavior of 26 calves? Did you use one camera for each calf? Please clarify. Also, behavior was measured for 30 minutes—when did this occur in relation to other measurements, and at what time of day?

It is not clear how the measurements were taken in terms of order. More clarification is needed regarding the sequence of data collection on the same day. Could you please provide more details on the order of measurements?

I kindly ask the authors to provide the data on the number of calves that lost their testicles at the end of the study, as well as the average day on which the testicles sloughed off.

I would also like to ask the authors about their observations of wound infection in both treatment groups. Could you please provide more details on this aspect of the study?

Finally, I ask the authors to carefully discuss the obtained data, specifically the behavioral data, while considering the limitations of the study.

Specific comments:

Please see the attached pdf.

Author Response

Responses to reviewer 2:

Reviewer 2: “The manuscript states that a modified version of a previously published scoring system was used. However, the scoring system presented in this manuscript appears to be identical to the originally published system. I suggest removing the table to avoid redundancy and potential copyright concerns, as it has been directly taken from the previously published manuscript (Marti et al 2017).”

Authors: Thank you for your comment and suggestion. We see the possible issue here, and will recite this, saying that the scoring system and score definitions came directly from Marti et al 2017. However, we believe that it is important to keep the table in the manuscript because we used our own pictures taking during the current study as examples of how we scored the wounds over the duration of the study.

Reviewer 2: “Another main concern is that individual housing may prevent a clear observation of certain behaviors. The stress or fear caused by isolation could influence the results and make it difficult to interpret the true effects of Lidoband on calf behavior. Could you clarify how the housing conditions were managed to minimize this potential impact? For example, I am not sure how you were able to observe the hind legs of animals housed in individual housing. Could you please clarify the methods used for this observation?”

Authors: Thank you for the comment and suggestion. Calves were housed indoors, individually, with panels (consisting of 8 bars) separating them, and no roof over the pens. They were still able to see one another completely, and neighboring calves could still touch each other. The main goal of the individual housing was to be able to monitor individual animal intake and behavior directed towards castration wound. On line 121, we will add a sentence to further explain the housing concept: “Sheep panels separated the individual pens, allowing calves to still see each other, as well as neighboring calves were still able to touch and interact with each other. There was no roof over the pens.”

Reviewer 2: “The authors mentioned that data from the daily human approach test were excluded from the data analysis due to calves being approached by humans before the actual test. Why is this information still included in the manuscript? Please delete any mention of the human approach test throughout the manuscript and adjust your objective accordingly. Given that the animals were individually housed, I don't think the Human Approach Test is the right method to measure the effect of lidoband on the affective states of calves, as individually housed animals are likely to be more fearful.”

Authors: Thank you for your comment. For further clarification, on line 237, it states that the human approach test was done daily for every study day -1 to 42 (43 days total). The days that were excluded were the days that the other data collections (body weights, blood collection, wound pictures) took place in the morning (days 0, 1, 2, 7, 14, 21, 28, 35, and 42) because of a substantial amount of human interaction. As addressed above, calves were only separated by panels, so they could still see each other, and neighboring calves could still touch each other. The authors find the mention of the human approach test valid, in a sense, that most of the time, all the calves approached the human.

Reviewer 2: “I noticed that you presented only the final body weight (BW) of the calves, but BW was measured at different time points. Could you explain why only the final BW was included in the manuscript? Additionally, did you ensure that the initial BW was not significantly different between the two groups? Please provide the average initial BW for both groups to facilitate comparison.”

Authors: Thank you for this comment. This is correct, that bodyweight was measured at multiple timepoints. However, on line 330, we state that there was no weekly treatment by time interaction for bodyweight (P>0.05), so we did not report further on this. We will report on line 321 the initial weight of both treatment groups, with the SEM and p-value. It will state “The average initial weight of the LLB group was 43.8 kg and the average initial weight of the CONT group was 46.5 kg (SEM 1.21, P=0.13).

Reviewer 2: “Please provide additional details on blood collection, specifically the type of blood collection tubes used and the timing of blood sample collection in relation to other measurements.”

Authors: Thank you for this comment. We will add the type of blood collection tubes for Cortisol (6mL Greiner Bio-One VACUETTE™ Lithium Heparin Blood Collection Tube, Greiner Bio-One North America Inc., Monroe, NC) and Substance P (6mL Greiner Bio-One VACUETTE™ Lithium Heparin Blood Collection Tube, Greiner Bio-One North America Inc., Monroe, NC) analysis on line 174. We will add on line 171 that blood was collected consistently at approximately 08:00 AM, after bodyweights, but before wound pictures.

Reviewer 2: “How did you use mounted cameras to record the behavior of 26 calves? Did you use one camera for each calf? Please clarify. Also, behavior was measured for 30 minutes—when did this occur in relation to other measurements, and at what time of day?”

Authors: Thank you for this comment. We will include on line 260 that cameras were mounted at a 45 degree angle on a rail that was 10 feet above the pens, with 2-3 calves able to be monitored per camera (Cameras 1, 2, 3, 4, 5 and 6 capture 3 pens per camera; Cameras 7, 8, 9 and 10 captured 2 claves per pen). Differences in number of animals per camera were due to the number of cameras we had available. Nevertheless, the bullet cameras allow for zoom and lens adjustment automatically using the Geovision program, that way only the pens of calves assigned to that specific camera would appear in the video. On line 268, we will also clarify that the 30-minute videos were rendered at approximately hour 08:30, after the morning feeding, as well as bodyweight, blood collection, wound pictures, and 15 minutes post human interaction.

Reviewer 2: “It is not clear how the measurements were taken in terms of order. More clarification is needed regarding the sequence of data collection on the same day. Could you please provide more details on the order of measurements?”

Authors: Thank you for the comment. To further clarify, I will add in the materials and methods, that data collection happened after the morning feeding, at approximately hour 08:00, in the order of body weight, blood collection, wound pictures, and video monitoring (15 minutes post human interaction).

Reviewer 2: “I kindly ask the authors to provide the data on the number of calves that lost their testicles at the end of the study, as well as the average day on which the testicles sloughed off.”

Authors: Thank you for this suggestion. In line 331, it states that 10/13 CONT calves and 8/13 LLB calves had cast their scrotal tissue by day 49. Due to not all the calves casting by day 49, we do not feel that it is appropriate to show the average day on which the testicles sloughed off, as the calves that still had not casted by day 49 would not be accounted for.

Reviewer 2: “I would also like to ask the authors about their observations of wound infection in both treatment groups. Could you please provide more details on this aspect of the study?”

Authors: Thank you for this comment. Wound infection, itself, was not recorded. However, in the wound scoring system, a score of 3 included purulent exudate, though there were no calves that showed concerning amounts, indicative of needing medical attention.

Reviewer 2: “Finally, I ask the authors to carefully discuss the obtained data, specifically the behavioral data, while considering the limitations of the study.”

Authors: Thank you for the suggestion. The authors agree the expanding on limitations of behavior data collection are beneficial to the discussion. The following paragraph was added to the discussion section.

 “Unfortunately, there is no set guidelines for times or intervals to score animal behavior. Other researchers assessing castration pain described their methodology of behavior scoring happening for 24 hours once a week using a 5-min interval scan sampling (Nogues et al., 2021). While Molony et al. (1995) behavior scored the first day for 3 hours using 2-minute intervals for the first 96 minutes and then 6-minute intervals for the remaining 84 minutes. In addition, when Meléndez et al. (2017) scored behaviors of calves submitted to castration, their strategy was to continuously behavior sampling between 2 and 4 hours post castration and then continuously scan sampling in 2-minute intervals every 10 minutes from 8:00 to 14:00 hours on days 1, 2 ,3, and 5 post castration using a subset of 5 animals per treatment. Later, Marti et al. (2017) used similar continuous scan sampling from 08:00 to 17:00 hours on days 5, 13, 20 and 27 post castration to evaluate chronic pain. Our strategy was to behavior score continuously for 30 minutes per day at the same time following the exact same scheduled on days (days -1, 0, 1, 2, 7, 14, 21, 28, 35, and 42), to maintain homogeneity and consistency of the video data. Videos were rendered after morning feeding, bodyweight measures, blood collections, and wound pictures were taken, then animals were allowed a period of 15 minutes to return to natural behaviors without human interference. Our goal to allow 15 minutes without human interaction was to diminish masking of pain behaviors from human observation since cattle can be stoic in the presence of humans. Nevertheless, the authors recognize that a 30-minute behavior scoring may not represent behaviors occurring during the 24 hours within a day. Behavior scoring present limitations due to amount of time necessary to score 24-hour videos, which is clear not only from the present study, but also other research groups previously mentioned (Molony et al., 1995; Nogues et al., 2021; Meléndez et al., 2017 and Marti et al., 2017). New artificial intelligence behavior tools to decrease the amount of video hours needed to be watched by a researcher may provide a better tool to address the current challenge.”

PDF specific comments:

Page 1:

 Reviewer 2: “Please specify the age of castration.”

Authors: Thank you for your comment, age was added to line 13.

Reviewer 2: “Please be specific. What type of performance are you referring to?”

Authors: Thank you for your comment, performance metrics were added to line 14.

Reviewer 2: “What do you mean by weekly performance? Please be specific.”

Authors: Thank you for your comment, this sentence has been removed, as we explain differences seen starting in the next sentence.

Reviewer 2: “How did standing and lying increase at the same time? This doesn’t make sense. Please check.”

Authors: Thank you for this comment. This will be changed on line 19 to “increased lying movements”.

Reviewer 2: “Please specify which clinical illness scores were measured in this study.”

Authors: Thank you for your comment, unfortunately the abstract only allows for a 200-word paragraph. That way it makes it difficult to include the data suggest here.

Page 2:

Reviewer 2: “Affective states are a broad category of feelings, moods, and emotions of animals. Please be specific and clearly state what particular aspects you focused on in this study.”

Authors: Thanks for your suggestion. On line 82, we will change the sentence to state “However, no studies compared welfare outcomes based on basic health and function, affective states (pain and discomfort), and natural living of calves castrated using lidocaine-infused bands for longer than 7 days.”

Reviewer 2: “Please see previous comment.”

Authors: Thank you. On line 82, we will change the sentence to state “The objectives of this study were to compare measures of welfare (basic health and function; and affective states (pain and discomfort) of calves…”

Reviewer 2: highlight

Authors: On line 88, this sentence will be changed to state “We hypothesized that calves banded with Lidoband ™ will show greater measures of basic health and function, decreased pain and discomfort, indicating benefits…”

Reviewer 2: “Could you clarify how the sample size for your study was calculated? Specifically, what variable or outcome measure was used for the power calculation?”

Authors: Thank you for your suggestion. We will add a sentence on line 93, stating “Due to the nature of a pilot study, sample size was determined by previous literature with similar study design [10,14].”

Reviewer 2: “Could you please elaborate further on the methods you used to ensure your study was blinded? Specifically, how did you prevent bias during data collection, analysis, and interpretation?”

Authors: Thank you for this suggestion. All calves were banded using a band with the same appearance (white rubber band), one contained lidocaine and the other contained no lidocaine. Bands were labeled as A and B, and a veterinarian not involved in data collection or analysis banded the calves at castration day. All researchers and animal caretakers were involved in data collection and data analysis were blinded to the treatment (LLB or CONT) until after data analysis were concluded.

The following information was described or added in the paper:

 On line 100, it states that the control group was banded with a standard band identical in appearance.

On line 130, it states that “All animal caretakers and analyzers were blinded to treatment group.”

We will add sentences on line 2 stating “Treatment groups were assigned as A and B at randomization by a researcher not involved in data collection or analysis. Researchers did not become unblinded to treatment group until all data analysis and results were completed.”

Reviewer 2: “Could you elaborate further on the randomization method used in your study?”

Authors: Thank you for your suggestion. On line 101, we will elaborate the sentence to say “Calves were randomized to treatment group A or B (to ensure blinding) by their study identification number, using a random number generator on Excel (version 3.12.1, Microsoft, Redmond, WA).”

Reviewer 2: “Could you explain why the study was followed specifically for 49 days? Was this duration chosen based on a specific rationale or study objective?”

Authors: Thank you for your suggestion. Forty-nine days was chosen in hopes of being able to follow all the calves through scrotal sloughing and the funding available. On line 109, we will reword the sentence to state “Banding occurred on October 11, 2023, which notes Day 0 of the study, and calves were then followed for 49 days, with the intention to observe scrotal sloughing for the majority of the calves.”

Page 3:

Reviewer 2: “Please specify the nutritional component of milk replacer used in the study.”

Authors: Thank you for this suggestion. This was added to line 125. (Herd Maker® PB Milk Replacer, Land O’Lakes®, Arden Hills, MN; 22% crude protein, 15% crude fat, 0.4% crude fiber)

Reviewer 2: “Please add the product name and the company of calf starter.”

Authors: Thank you for this suggestion. This was added to line 127. (Calf Grower B-68, Mid Kansas Coop, Manhattan, KS); 14% crude protein, 2% crude fat, 7% crude fiber)

Reviewer 2: “This is repetition please delete it.”

Authors: Thank you for this suggestion, we have removed the information about the scale on line 132.

Reviewer 2: “Please provide additional details on how this ratio was calculated.”

Authors: Thank you for this suggestion. The sentence on line 139 has been reworded to state “With daily feed and milk intake measured on a dry matter basis, as well as weekly body weights, a gain to feed ratio (G:F) was calculated to determine feed conversion by dividing the weight gained by the feed consumed.”

Reviewer 2: “Could you provide more details on how the wound pictures were taken? Specifically, what distance was maintained, and what type of camera or equipment was used?”

Authors: Thank you for this suggestion. We will add to the sentence on line 143 to state “Using a handheld camera (Canon EOS Rebel T3i, Ohta-ku, Tokyo, Japan), photographs were taken approximately 30 cm from the caudal end of the calves while they were standing, with the entirety of the scrotal tissue and band visible.”

Reviewer 2: “What do you mean by this sentence. Please clarify.”

Authors: Thank you for this suggestion. We have reworded this sentence on line 149 to state “While checking for band retention, date of scrotal sloughing was recorded, if applicable.”

Reviewer 2: “How you measure the restriction of blood flow?”

Authors: Thank you for this question. While blood flow was not directly measured in this study, it was assumed that after the band was placed around the scrotum, that the blood flow was restricted, allowing the scrotum and testicle tissue to die. How banded castration works is described on lines 60-64. Score 1 definition was changed to “band placed, restricting blood flow, but scrotum is not necrotic”

Page 4:

Reviewer 2: “Please provide additional details on blood collection, specifically the type of blood collection tubes used and the timing of blood sample collection in relation to other measurements.”

Authors: Thank you for this comment. We will add the type of blood collection tubes for Cortisol and Substance P analysis on line 176. We will add on line 174 that blood was collected at approximately hour 08:00, after feeding, bodyweights, and before wound pictures were collected.

Page 5:

Reviewer 2: “Please provide more information on how you prepared the benzamine at the required concentration and how you ensured the tubes were kept clean during the addition process.”

Authors: Thank you for this suggestion. This will be reworded to state “The Benzamidine HCl solution was prepared in a clean lab wearing proper PPE including gloves and a lab coat. Briefly, 156.6mg Benzamidine HCl was weighed out and added to 10mL MilliQ ultrapure water in a sterile 15mL conical. Solution was vortexed thoroughly and until dissolved. Blood tube caps were removed briefly to deliver the 60uL of Benzamidine HCL solution to each tube using an Eppendorf M4 Repeat pipet. Tubes were recapped immediately and stored at 4°C until use within 48hrs..”

Reviewer 2: “Given that the animals were individually housed, I don't think the Human Approach Test is the right method to measure the effect of lidoban on the affective states of calves, as individually housed animals are likely to be more fearful”

Authors: Thank you for this comment. Calves were housed indoors, individually, with panels (consisting of 8 bars) separating them, and no roof over the pens. They were still able to see one another completely, and neighboring calves could still touch each other. The main goal of the individual housing was to be able to monitor individual animal intake and behavior directed towards castration wound. On line 121, we will add a sentence to further explain the housing concept: “Sheep panels separated the individual pens, so calves were still able to see each other, as well as neighboring calves were still able to touch and interact with each other. There was no roof over the pens. Nevertheless, due to human exposure daily calves approached the human almost every time the human approach was performed.

Reviewer 2: “This is not clear. Did you place tag cement on the animal's skin to secure the hobo? Please clarify.”

Authors: Thank you for this suggestion. Yes, the small dab of tag cement was placed on the hair of the leg of the animal. We will reword the sentence on line 243 to state “To attach the accelerometer, a small dab of tag cement (Kamar Adhesive) was placed on the hair of the calves leg and four -inch cohesive bandage…”

Page 6:

Reviewer 2: “How did you use mounted cameras to record the behavior of 26 calves? Did you use one camera for each calf? Please clarify. Also, behavior was measured for 30 minutes—when did this occur in relation to other measurements, and at what time of day?”

Authors: Thank you for this suggestion. I addressed this in my reply to your comment above: We will include on line 260 that cameras were mounted on a rail that was 10 feet above the pens, with 2-3 calves able to be monitored per camera. On line 268, I will also clarify that the 30-minute videos were rendered at approximately hour 08:30, after the morning feeding, as well as bodyweight, blood collection, and wound pictures.

Reviewer 2: “I am not sure how you were able to observe the hind legs of animals housed in individual housing. Could you please clarify the methods used for this observation?”

Authors: Thank you for the question, we have addressed the individual housing in the previous questions. Calves were housed indoors, individually, with panels (consisting of 8 bars) separating them, and no roof over the pens. They were still able to see one another completely, and neighboring calves could still touch each other. All limbs were seen on video at all times. The main goal of the individual housing was to be able to monitor individual animal intake. On line 121, we will add a sentence to further explain the housing concept: “Sheep panels separated the individual pens, so calves were still able to see each other, as well as neighboring calves were still able to touch and interact with each other. There was no roof over the pens.”

Reviewer 2: “So, you excluded the daily human approach test from the data analysis due to calves being approached by humans. Why is this information still included in the manuscript? Please delete any mention of the human approach test throughout the manuscript and adjust your objective accordingly.”

Authors: Thank you for the question, we have addressed the individual housing in the previous comments. For further clarification, on line 237, it states that the human approach test was done daily for every study day -1 to 42 (43 days total). The days that were excluded were the days that the other collections (body weights, blood collection, wound pictures) happened in the morning (days 0, 1, 2, 7, 14, 21, 28, 35, and 42) because of a substantial amount of human interaction. As addressed above, calves were only separated by panels, so they could still see each other, and neighboring calves could still touch each other.

Reviewer 2: “This is not defined in the Materials and Methods. Please define what you mean by "wound granulation."”

Authors: Thank you for this suggestion. On line 288, we added a sentence stating “Wound granulation in the current study is defined as the occurrence of new tissue, pink or red in color, formed on the surface of the scrotal wound during the healing process.”

Reviewer 2: “So, this means that you ended the study early before the complete healing of the castration wound. Please add this to the study limitations.”

Authors: Thank you for this suggestion. Unfortunately, due facilities and funding constraints we were unable to follow those calves for a longer period of time. A sentence was added to line 449 stating “A limitation to the current study is the fact that not all calves sloughed their scrotal tissue and had completely healed wounds by day 49. Further research is warranted to follow all banded calves through the time of scrotal sloughing and complete wound healing.”

Reviewer 2: “Please explain in more detail how the data were transformed and back-transformed.”

Authors:  Data transformation is explained on materials and methods, data analysis section, line 296. “The percentage of time spent standing while not eating, percent time spent standing while eating, and percent time spent lying were square root +1 transformed to achieve normality. Counts of tail flicking, looking at flank, and wound licking were log10 +1 transformed to achieve normality.” We have not back transformed values to present the data as stated on our figures and tables. That was done to follow similar published behavior research. For clarity we added the following sentence on line 299 “Behavior data was transformed for statistical analysis and are presented as least squares means and standard error without transformation, with P-value from transformed values” to the statistical analysis.

Reviewer 2: “What do you mean by "overall performance"? Please be consistent throughout the manuscript. Why were time points not included as a fixed effect in the growth performance data?”

Authors: Overall performance is only comparing between treatment groups bodyweight (Final Weight), ADG (Final ADG) and G:F (Final G:F) at day 49 of the trial. Reporting the data as overall performance is a common practice in nutrition and performance animal trials and that describes if in a longer period of time the effects of the treatment carryover. That way no repeated measures are accounted for the overall the time cannot be included in the model. Nevertheless, we have analyzed the data weekly, and the data is presented in the text within Results 3.1. The weekly data represents the repeated measures overtime. We will reword this sentence on line 303 for clarification, stating “For the models accounting for overall performance (final bodyweight at day 49 post castration, and overall ADG and G:F over the 49 days of trial), fixed effects included…”

Reviewer 2: “Did you need to replace the accelerometer throughout the study? When did this happen, and how did it affect data collection? Please provide this information in materials and methods.”

Authors: Thank you for this question and suggestion. Yes, accelerometers did need to be replaced throughout the study due to storage capabilities (line 307). Accelerometers were replaced during data collection timepoints, while the calves were laying down and restrained for blood collection. No data was lost since loggers were launched on the day of the replacement at midnight and times of removal (old logger) and replacement (new logger) were recorded. Data was the stitched together to the minute of logger swap with no data loss. A sentence was added to line 248 stating “Due to limited storage capacities, accelerometers were replaced twice over the duration of the study. Logger replacements happened on collection days, while calves were laying down and restrained for blood collection. Time of removal and reposition were recorded, since loggers were launched for the day of replacement at midnight it allowed the data to be stitched together without data loss.”

Reviewer 2: “You mentioned previously that the human approach test data were excluded from the analyses. Why is it included here?”

Authors: Thank you for the question, we have addressed the individual housing in the previous questions. For further clarification, on line 237, it states that the human approach test was done daily for every study day -1 to 42 (43 days total). The days that were excluded were the days that the other collections (body weights, blood collection, wound pictures) happened in the morning (days 0, 1, 2, 7, 14, 21, 28, 35, and 42) because of a substantial amount of human interaction. As addressed above, calves were only separated by panels, so they could still see each other, and neighboring calves could still touch each other.

Page 7:

Reviewer 2: “Why did you only present the final body weight, given that you measured body weight at different time points? Did you ensure that the initial body weight was not significantly different before the start of the study? Please provide the average initial body weight for both groups.”

Authors: Thank you for the question. This is correct, that bodyweights were measured at multiple timepoints. However, on line 330, we state that there was no weekly treatment by time interaction for overall bodyweight (P>0.05), so we did not report further on this. Homogeneity of the body weights were in fact tested and we will report on line 321 the initial weight of both treatment groups, with the SEM and p-value. “The average initial weight of the LLB group was 43.8 kg and the average initial weight of the CONT group was 46.5 kg (SEM 1.21, P=0.13).”

The authors would like to thank reviewer 2 for the questions and suggestions that helped improve the description and overall understanding of our trial. Below is a graphic of the pen layout to help explain how the calves were not completely separated from each other and how the cameras were able to show multiple calves. Another picture shows an example from the camera view.

Reviewer 4 Report

Comments and Suggestions for Authors

Line 13-15, This sentence should be modified. The main sentence states that LLB is comparable to standard band. However, your next sentence then lists how it is better. I would delete comparable, if not outperformed, and just state that LLB was better than standard band.

Line 39, I would say that beef x dairy is a common practice. It can not be constant growth as in long run dairy herd becomes smaller. In another 2-3 years we will be at max adoption of beef on dairy crosses.

Line 52, maybe say avoid pregnancies in heifers

Line 54, I would say place band around the scrotum not above, maybe around the scrotum above the testicles.

Line 58, compared to mostly. Although I don’t know that you have to compare to surgical castration. I would be inclined to end the sentence at “acute and chronic pain involved.

Line 67, define EC50 and EC95

Line 198, was time of day standard? What time of day was observations done? Observation compared to other events such as feeding, sunrise or sunset, etc?

Line 233, include weight range. Did you do any stats to confirm that there were no differences between treatment groups regarding weight, age, etc. if so that should be reported.

Line 306, did you do a power calculation?

Author Response

Responses to reviewer 3:

Reviewer 3: “Line 13-15, This sentence should be modified. The main sentence states that LLB is comparable to standard band. However, your next sentence then lists how it is better. I would delete comparable, if not outperformed, and just state that LLB was better than standard band.”

Authors: Thank you for this suggestion. The first sentence states that LLB was comparable, if not, outperformed, the standard band. We will delete the second sentence, on line 16, because the following sentence starts explaining the differences seen between the two treatment groups.

Reviewer 3: “Line 39, I would say that beef x dairy is a common practice. It can not be constant growth as in long run dairy herd becomes smaller. In another 2-3 years we will be at max adoption of beef on dairy crosses.”

Authors: Thank you for this suggestion. We will change the sentence on line 39 to state “The practice of crossbreeding beef x dairy cattle has grown significantly in the U.S.”

Reviewer 3: “Line 52, maybe say avoid pregnancies in heifers”

Authors: Thank you for this suggestion. We will change this sentence on line 57 to state “avoid pregnancies in cattle”.

Reviewer 3: “Line 54, I would say place band around the scrotum not above, maybe around the scrotum above the testicles.”

Authors: Thank you for this suggestion. We will restate this sentence on line 60 to say “Banding consists of placing a latex rubber band around the scrotum, entrapping the spermatic cords…”

Reviewer 3: “Line 58, compared to mostly. Although I don’t know that you have to compare to surgical castration. I would be inclined to end the sentence at “acute and chronic pain involved.”

Authors: Thank you for this suggestion. We will restate this sentence on line 63 as “Significant welfare concerns are pain and distress associated with this procedure due to the acute and chronic pain.”

Reviewer 3: “Line 67, define EC50 and EC95”

Authors: Thank you for this suggestion. We will restate the sentence as “It has been shown that calves banded with a lidocaine-infused castration band had tissue levels of lidocaine that approached the 50% effective concentration (EC50) and exceeded the 95% effective concentration (EC95) at 2 and 72 hours, respectively.”

Reviewer 3: “Line 198, was time of day standard? What time of day was observations done? Observation compared to other events such as feeding, sunrise or sunset, etc?”

Authors: Thank you for the comment. To further clarify, I will add in the materials and methods, on line 133, that data collection happened after the morning feeding, at approximately hour 08:00, in the order of body weight, blood collection, wound pictures, and video monitoring (15 minutes post human interaction).

Reviewer: “Line 233, include weight range. Did you do any stats to confirm that there were no differences between treatment groups regarding weight, age, etc. if so that should be reported.”

Authors: Thank you for this suggestion. We will report on line 321 the initial weight of both treatment groups, with the SEM and p-value. It will state “The average initial weight of the LLB group was 43.8 kg and the average initial weight of the CONT group was 46.5 kg (SEM 1.21, P=0.13).”

Reviewer: “Line 306, did you do a power calculation?”

Authors: Thank you for your suggestion. We will add a sentence on line 93, stating “Due to the nature of a pilot study, sample size was determined by previous literature with similar study design [10,14].”

The authors would like to thank Reviewer 3 for all the suggestions and questions to improve our manuscript.